# Use of Anaerobic Digestate Inoculated with Fungi as a Soil Amendment for Soil Remediation: A Systematic Review

**DOI:** 10.3390/biology14111579

**Published:** 2025-11-11

**Authors:** Mónica López Velarde Santos, José Alberto Rodríguez Morales, Yesenia Mendoza-Burguete, María del Carmen González-López, Héctor Pool, Aldo Amaro-Reyes, Juan Campos-Guillén, Miguel Angel Ramos-López, Carlos Eduardo Zavala Gómez, Ricardo Chaparro-Sánchez

**Affiliations:** 1Facultad de Ingeniería, Universidad Autónoma de Querétaro, Cerro de las Campanas S/N, Las Campanas, Querétaro 76010, Mexico; monica.lopez.velarde.santos@uaq.mx (M.L.V.S.);; 2Facultad de Química, Universidad Autónoma de Querétaro, Av. Panamericana 180 int. 100, Centro 7670 Pedro Escobedo, Querétaro 76010, Mexico; 3Facultad de Química, Universidad Autónoma de Querétaro, Cerro de las Campanas S/N, Las Campanas, Querétaro 76010, Mexico; 4Facultad de Informática, Universidad Autónoma de Querétaro, Querétaro 76010, Mexico

**Keywords:** soil amendment, mycorrhizal, fungi, anaerobic digestate, soil remediation, contaminants

## Abstract

This systematic review gathers important information regarding the use of anaerobic digestates inoculated with fungi, considering their use as a soil amendment to enhance contaminant removal, plant growth, and soil stabilization. Notably, the excessive accumulation of nutrients contained in digestates, such as phosphorous and nitrogen, could have negative impacts on ecosystems; furthermore, contaminants—such as those derived from mining or industrial activities—represent a latent risk for animals and humans if they enter the food chain. Both these problems could be addressed by inoculating the anaerobic digestates used for soil remediation with fungi. For this review, a systematic search was performed to retrieve relevant scientific studies published in the past ten years. The combined application of anaerobic digestates and fungi contributes to contaminant reduction while improving soil structure. Important findings include that the biomass source used for anaerobic digestion plays a crucial role, with the sole use of cattle manure achieving the best results in terms of fungal colonization and organic matter degradation. It has also been suggested that optimal temperatures result in higher remediation efficiencies. Further research should focus on determining the specific vegetation, existing contaminants, and fungal strains to be used.

## 1. Introduction

Severe environmental problems have arisen with the growth and development of human populations, including climate change; global warming; sea level elevation; soil, water, and air pollution; biodiversity loss; overharvesting; and deforestation. The consequences of environmental damage affect human beings, either directly or indirectly [1], with economic impacts in the agricultural, fishing, and tourism industries having been observed [2]. Pollution can lead to public health problems such as respiratory illnesses or waterborne diseases [3]. The incidence of problems such as resource scarcity, food chain disruptions, and environmental disasters is increasing worldwide. A multidisciplinary approach should be developed and implemented to overcome these problems, particularly focusing on the mitigation of greenhouse gas emissions, use of renewable energies, pollution reduction, and remediation of natural habitats.

Anaerobic digestion (AD) has been widely used for the simultaneous production of green energy and waste management. Aside from biogases, anaerobic digestate is also obtained as a result after the AD process. This digestate is rich in nutrients that are essential for plant growth and, thus, can be used as a fertilizer [4].

Mycoremediation refers to the use of fungi to remove or break down pollutants present in soil and water. Fungi—e.g., mycorrhizae or endophytic fungi—have the capacity to degrade pollutants such as heavy metals or pesticides, reducing their harmful effects on the environment [5].

Several scientific publications have focused on the sole use of anaerobic digestates as soil amendments or mycoremediation alone for soil pollutant removal [4,5,6,7,8,9,10,11]; however, few studies regarding the simultaneous use of both can be found. These technologies show promise as more effective and cost-efficient approaches for reducing or removing pollutants from soils when combined than individually, showing beneficial impacts on soil fertility and plant growth.

Although digestates could be used to promote better fungal growth and, thus, more efficient pollutant removal, few studies on this topic have been published to date. This systematic review aims to describe the anaerobic digestion process, the generated byproducts (e.g., digestates), and their use for contaminant removal. Additionally, mycoremediation is described, as well as the inoculation of digestates with fungi for improvement of their contaminant removal and plant growth effects. The main goal of this review is to analyze the effects of mycoremediation with digestates on pollutant removal, plant growth, and soil stabilization. The obtained results can contribute to developing innovative and cost-effective alternatives for soil management and contaminant removal.

### 1.1. Anaerobic Digestion Process

AD involves a series of biochemical reactions in which different bacterial consortia break down organic matter into its individual components, forming a mixture of gases called biogas [12]. Biogas is a byproduct of microbial metabolism, with different microorganisms having the ability to convert organic matter almost entirely into biogas [13,14]. In principle, all types of organic matter can be degraded under anaerobic conditions for biogas production, including poultry droppings, agricultural crop wastes, or cattle manure [13]. However, not all organic matter components can be broken down by the same bacterial strains. Furthermore, biomass with high woody substance content decomposes slowly due to the presence of lignin. The organic material used to produce biogas is called the substrate, the chemical composition of which—particularly in terms of carbohydrates, fats, and protein content—affects the amount of biogas produced and its methane content.

As shown in Figure 1, the process of biogas production is divided in four steps: Hydrolysis, acidification (acidogenesis), acetic acid formation (acetogenesis), and methane formation (methanogenesis) [14,15,16]. During hydrolysis, the complex compounds in the biomass with high molecular weight (e.g., lipids, carbohydrates, and proteins) are broken down into smaller, more soluble organic compounds (e.g., fatty acids, glucose, and amino acids) [15]. In the second step, acidification, acid-forming bacteria break down the intermediate products into smaller fatty acids (e.g., acetic, propionic, and butyric acid), as well as carbon dioxide and hydrogen. During the third step, acetogenesis, products derived from the second step are then converted by acetogenic bacteria into biogas precursors such as acetic acid, hydrogen, and carbon dioxide. In this step, water content can negatively affect the process, and an excessively high hydrogen content prevents the conversion of intermediate products for energetic reasons. Products which inhibit methane formation may also form during this step, such as organic acids, propionic acid, isobutyric acid, isovaleric acid, and caproic acid. For this reason, the acetogenic bacteria (hydrogen producers) must be in close association with hydrogen-consuming methanogenic archaea, which consume hydrogen together with carbon dioxide in order to form methane, thus ensuring optimal environmental conditions for the growth and reproduction of acetic acid-producing bacteria [14]. During the methanogenesis step, acetic acid, hydrogen, and carbon dioxide are mainly converted into methane by strictly anaerobic methanogenic archaea. Hydrogenotrophic methanogens produce methane from hydrogen and carbon dioxide, whereas acetoclastic methanogens form methane through the division of acetic acid; in particular, around 70% of methane derives from acetic acid decomposition, while only 30% is produced through hydrogen utilization [14,17].

Different bacterial strains multiply at different speeds, with the doubling time defined as the time it takes for a bacterial population to double in size [18]. The bacterial strains utilized in the first two steps—hydrolysis and acidogenesis—have a remarkably slower doubling time than methanogenic archaea. The doubling times of the bacteria/archaea strains involved in biogas production are shown in Table 1 [18].

Key factors affecting the well-being of different bacterial strains—and, thus, the efficiency of anaerobic digestion processes—should be studied in depth. Factors to consider inside the bioreactor include the temperature, oxygen level, pH value, nutrient supply, and the presence of inhibitory substances. Operating parameters that should be monitored during the AD process include the hydraulic retention time (i.e., the time the substrate remains in the bioreactor; d), organic matter loading rate (kg · m^−3^ · d^−1^), CH_4_ productivity (m^3^ CH_4_ · m^−^^3^ · d^−1^), CH_4_ yield (m^3^ · t^−1^), biomass degradability (%), and the mixing of substrates in the bioreactor. According to Fachagentur Nachwachsende Rohstoff [14], although the composition of biogas is influenced by the process control parameters to some extent, it primarily depends on the composition of the input material.

### 1.2. Biogas Composition

The composition of biomass may vary but, in general, it contains methane (CH_4_), carbon dioxide (CO_2_), water (H_2_O), hydrogen sulfide (H_2_S), nitrogen (N_2_), and hydrogen (H_2_), as detailed in Table 2 [19].

Regarding the quality of the gas mixture, the concentration of hydrogen sulfide (H_2_S) plays an important role. H_2_S is found in biogas as a trace gas in very small amounts, as shown in Table 2, which can lead to issues for two key reasons. First, its level should not be too high, as even low concentrations of this gas can inhibit the degradation process. Second, high H_2_S concentrations in biogas can lead to corrosion damage in combined heat and power plants and boilers during use [20,21].

The methane content in biogas is of primary importance, as it represents the combustible portion of biogas and, thus, directly influences its calorific value. The achievable yield of methane is essentially determined by the composition of the substrate used, mainly in terms of the proportion of fats, proteins, and carbohydrates. The specific methane yields of the aforementioned groups of substances decrease in the mentioned order, such that a higher methane yield can be achieved with fats than with carbohydrates [22]. Table 3 lists the specific biogas yields and methane contents of the corresponding substance groups [23].

Few biogas plants practice mono-digestion; namely, the digestion of only one kind of substrate. Most anaerobic digesters use different crops or seasonal substrates, which guarantees the effectiveness of the AD process, increasing the digester loading and methane production capacities. This ameliorates the buffer capacity, lowers the pH level in the methanogenesis step, and contributes to a better nutrient balance (e.g., carbon/nitrogen ratio; C/N), among other improvements [21]. Baştabak and Koçar [24] have suggested that the hydraulic retention time (HRT) and temperature in the bioreactor are interconnected: as temperature increases, HRT decreases. Both of these parameters are very important for pathogen inactivation.

### 1.3. Use of Biogas

The produced biogas can directly be used as a cooking fuel or injected into a co-generator (CHP) for simultaneous heat and power generation. Additionally, the CH_4_ in biogas can be enriched to upgrade the biogas to biomethane, such that it can be used with minimal modifications as natural gas. Finally, biogas can be used to produce value-added chemicals which are used in energy or industrial processes [13,17,21,25]. Figure 2, modified from Mertins and Wawer [25], shows the different pathways for biogas utilization.

Biogas is thus a clean energy source, contributing to a reduction in greenhouse gas emissions. Kabeyi and Olanrewaju [13] have reported that, in Switzerland, 8% of renewable energies are produced from biogas. The use of biogas can also diminish the use of firewood for cooking. It has been reported, that about 200 million people will use biogas as cooking biofuel in Asia and Africa by 2040, contributing to the social development of emerging and developing countries [13,21]. In 2018, it was reported that China was the biggest producer of biogas worldwide, with around 50 million small-scale bioreactors as well as 4000 farm-scale and 2500 industrial-scale reactors. India has also been reported to have many small-scale biogas plants, and Europe has shown an increasing trend in biogas production [21]. In 2022, Zupancic et al. [17] recorded around 132,000 biogas plants worldwide, of which 17,783 with an installed capacity of 10.5 GW were located in European countries. Furthermore, 700 biogas plants for biomethane upgrading were reported, of which almost 80% were located in Europe. After 2005, the number of biogas plants increased tremendously, especially in Germany, France, Switzerland, and Holland [17].

Biowastes, such as sludges, manures, or agricultural residues, are being dispersed in agricultural soils as biofertilizers or landfilled. The continual disposal of residues in soils can lead to the accumulation of nutrients such as nitrogen, phosphorous, or potassium, as well as heavy metals, leading to negative health impacts in pasture-raised cattle [17]. In addition, the disposal of biowastes in soils may serve as a significant contamination source, whereas landfill gases consisting of volatile organic compounds, CH_4_, and CO_2_ are released into the environment indiscriminately. Landfill gases have been shown to have a great impact on ozone layer depletion [17].

AD could be successfully used for the optimal conversion of biowastes to bioenergy in the form of biogas, while simultaneously obtaining biofertilizer as a byproduct. After biogas production, digestates remain as residue, which is a partially stabilized wet suspension of both available and less degradable organic matter, along with minerals and microorganisms. Digestate still contains an important amount of organic matter with residual biogas potential, which can be further decomposed or processed into valuable products such as biofertilizer for agricultural soils [17]. As a biofertilizer, digestate provides nutrients which are necessary for humus and soil structure maintenance, thus promoting crop growth [21].

### 1.4. Anaerobic Digestate

As noted above, biogas digestate is rich in nutrients which are beneficial for plant growth. The nutrient content and properties depend on the input material and operational parameters of the biogas plant. Crucial factors affecting digestate quality include bacterial activity, water and nutrient contents, the C/N ratio in input material, particle size and concentration, inhibitory or toxic compounds, pH, presence of oxygen, microbial composition, and reactor parameters such as temperature, design, and mixing [26,27,28]. Several authors have reported that biogas digestate is a valuable product which can be used as a biopesticide, fertilizer, animal feed, or even for the obtention of irrigation water or phosphate salt [29,30,31,32,33,34]. Figure 3 shows the possible uses of biogas digestate, according to Baştabak and Koçar [24].

When using digestate as a biofertilizer, the biostability of the digestate should be taken into consideration; in particular, the digestate should not include pathogens or a high organic matter content, thus ensuring that it is not hazardous to living things. The main factor influencing the biostability of digestate is the bioreactor temperature. After AD, the mass content decreases by 90–95%. In addition, when comparing digestate with cattle manure, pathogenic populations of *E. coli* and *Salmonella* sp. were present in manure, whereas digestate obtained at mesophilic temperatures showed almost no pathogens [35]. The only disadvantage of sanitization is the loss of N in ammonia, which can be overcome by adding ammonium sulfate to the digestate [24].

As previously mentioned, the AD substrate affects directly the biogas production yield as well as the digestate’s characteristics. When digesting urban residues, the amount of heavy metals in digestate could reach toxic concentrations; therefore, its use as a biofertilizer without any treatment is not recommended. According to Jasinska et al. [36], dry matter, organic matter, pH, N, P, Cd, Cu, Ni, Pb, Zn, Hg, and Cr levels should be analyzed in anaerobic digestate at least every six months. Limit values for six key trace elements (TEs) in anaerobic digestates for agricultural use (e.g., as fertilizer) are shown in Table 4 [36].

The nutrient composition of anaerobic digestates derived from organic wastes vary according to the waste’s constitution; they are usually rich in P, N (in form of NH_4_^+^), and/or Na^+^ and Cl^−^ ions, among others. A high NH_4_^+^ content could cause ammonium toxicity in the anaerobic digestion process. High concentrations of Na^+^ and Cl^−^ ions—and, thus, NaCl—have detrimental effects on plant growth, reducing water and nutrient uptake. In addition, excessive concentrations of Na^+^, Cl^−^, P, or N can have toxic effects on plant growth [37]. As a biofertilizer, the use of anaerobic digestate can increase crop yields; nevertheless, there are still many challenges related to its efficient use.

To avoid this problem, the associated risk might be minimized by reducing contaminant factors through denitrification applications, ammonia stripping, adsorption, membrane filtration/concentration, struvite crystallization/precipitation, and evaporation [24,37]. Furthermore, remediation techniques could be applied to diminish the concentrations of toxic compounds in digestates, thus enhancing crop growth. Remediation utilizing fungi—a process called mycoremediation—could be considered as an efficient alternative. Anaerobic digestate could be applied in soils, where mycorrhizae colonize plant roots for contaminant removal; in this way, better soil nutrient adsorption and, thus, plant growth could be promoted.

### 1.5. Mycoremediation

Mycoremediation is a remediation method in which fungi are used to degrade or remove pollutants from ecosystems. Mycorrhiza refers to the mutualistic associations between plant roots and mycorrhizal fungi. Mycorrhizal fungal hyphae, which make up the body of a mycorrhizal fungus, are fixed to plant roots and soil particles and build filaments which tolerate contaminants such as heavy metals, leading to better ability of the plant to adapt to nutrient diminishment, as well as temperature or pH modifications [38]. Mycorrhizal fungi are found in the roots of many plants. These symbiotic associations benefit the host plants through enhanced water or soil nutrient uptake, particularly P and N. Furthermore, mycorrhizal fungi collaborate with other soil microorganisms, further enhancing nutrient absorption [39,40]. The structure and function of mycorrhizal associations are wide, including arbuscular mycorrhiza (AM), ectomycorrhiza (EcM), ectendomycorrhiza, arbutoid mycorrhiza, monotropoid mycorrhiza, ericoid mycorrhiza, and orchid mycorrhiza, among which AM and EcM are the most common; moreover, AM fungi are economically and ecologically important. AM fungi are used as biofertilizers and cannot be cultivated without a host plant. The intracellular hyphal network of EcM fungi shows more extensive associations than Arbuscular Mycorrhizal Fungi (AMFs) [39]. AMFs can either increase heavy metal (HM) uptake and transport from roots to shoot, or stabilize HMs in roots and shoots, diminishing their uptake. Some authors have reported that AMFs enabled the accumulation of HMs in a non-toxic form within the roots of the plant and the extracellular mycelia [39].

#### 1.5.1. Effects of HMs in Soils

The presence of HMs in soils could create hazards for both plants and consumers, especially when found over safe and tolerable limits. High concentrations of HMs have negative effects on microorganisms and microbial processes, leading to reductions in plant growth. HMs can enter soils in different ways, such as fossil fuels, mining and smelting, municipal wastes, and the use of fertilizers or pesticides [41,42,43]. Furthermore, while some metals such as Zn, Cu, Mn, Ni, and Co are essential for plant growth in trace amounts and are considered micronutrients, other metals such as Cd, Pb, or Hg do not support any biological functions [43]. Excessive HM concentrations can induce metabolic disruptions in plants, altering cellular activities and affecting nutrient uptake, plant development, or even inducing reactive oxygen species production. This could be reflected in poor plant growth, a reduction in turgor stem pressure, leaf chlorosis, decreased seed germination, and senescence [41]. Metals can be found in different forms in soils, such as free ions, soluble complexes, exchangeable ions, precipitated or insoluble oxides, carbonates and hydroxides, or even as silicate materials. The toxic effects of metals depend on their ability to be transferred from the soil to living organisms—namely, their bioavailability—which depends on both physicochemical and biological factors [43].

#### 1.5.2. Mycorrhizae for Metal Remediation

Mycoremediation refers to degrading or removing contaminants from the environment using fungi cultivated in or on a host plant, which is an economic and effective alternative to reduce the concentrations of soil contaminants. Metal-tolerant plants have been used to extract HMs from soils, which translocate and accumulate HMs from the soil into their shoots, leaves, and other structures. Through colonizing these plants with mycorrhizal fungi, greater plant growth, protective mechanisms, and buffer capacity for abiotic stresses can be achieved. Several studies have reported increased metal uptake in plants inoculated with AMFs, especially regarding As, Cr, Cd, Pb, Zn, Mn, Cu, Al, and Co. Thereby, increased metal contents in the roots, shoots, and fronds were observed, allowing phytoextraction to more easily take place [39,41].

In soils that are heavily contaminated with HMs, plant colonization through mycorrhiza has been observed. Several authors have reported high levels of mycorrhizal colonization in agricultural soils polluted with heavy metals [43].

Plant AMF tolerance may occur due to different factors. On one hand, higher nutrient (especially P) uptake leads to greater plant growth and, thus, a higher availability of biomass for metal distribution within the plant. On the other hand, AMF-assisted plants have a better capacity to bioaccumulate metals, hindering their translocation to shoots and roots [41,44,45].

Overall, mycoremediation could be used as a low-cost alternative for soil remediation which shows beneficial effects on plant growth, metal attenuation, and productivity.

### 1.6. Use of Anaerobic Digestate in Mycoremediation

According to [6], digestate application in the crop production context led to a yield increase between 15% and 28%. The successful use of digestate in horticulture has been reported in [7]. In Caracciolo et al. [8] the microbial community was studied after the addition of manure digestate, which was indicated as a successful strategy for soil restoration. The results of Slepetiene et al. [9] indicated that the use of digestate is more efficient immediately after its addition to soil, due to the fact that the nutrient content to support plant growth decreases over time. To augment the potential of digestate and prevent nutrient loses, digestate addition should be performed in accordance with specific plant growth stages.

Nevertheless, the optimization of anaerobic digestion processes typically focuses only on enhancing biogas production, without taking into consideration the toxicity of the digestate. While the successful use of digestate as a biofertilizer has been reported, legal regulations on digestate management should be considered in order to ensure that it is used in a safe manner; for example, considering hygienic standards, pathogen elimination, fermentation degree, odor, organic matter content, heavy metal content, or even seed elimination [10].

The effects of mycoremediation on plant growth and contaminant removal have been tested in many scientific studies. In the results of Kumar and Dwivedi [11], it was reported that mycoremediation could serve as a sustainable alternative to treat HM-polluted soils. It was elucidated that fungi have the potential to remediate soils polluted with organic, inorganic, and emerging contaminants, such as HMs, pharmaceuticals, polycyclic aromatic hydrocarbons (PAHs), pesticides, and herbicides. Pesticide degradation rates of 91%, 94%, and 98.4% were achieved with the fungal strains *Pleurotusdryinus*, *Trameteshirsuta*, and *Aspergillusniger*, respectively. Akhtar [5] concluded that mycoremediation could be used as a cost-effective strategy for the removal of recalcitrant pollutants, such as antibiotics, herbicides, polyaromatic hydrocarbons, insecticides, drugs, cyanotoxins, detergents, HMs, or even plastics.

In this sense, combination of the technologies of mycoremediation and digestate application in soils is posed to be a cost-effective and efficient alternative to break down or remove pollutants. Simultaneously, beneficial impacts for soil fertility and organic carbon contents are expected, leading to more efficient plant growth [46]. Furthermore, the application of biogas production-derived digestate along with higher fungi or mycorrhizal root colonization could provide augmented symbiotic benefits for plant development—even in polluted soils, as the plants can better adapt to the associated stresses [29,36,47,48,49,50,51,52].

### 1.7. Theoretical Basis

Although the information gathered in different platforms is helpful, the use of appropriate keywords to select studies addressing a specific topic from a large number of papers can result in a more efficient search methodology, leading to more accurate results. This systematic review aims to understand the effects of the application of digestates inoculated with fungi in the context of contaminant removal, soil stabilization, and plant growth. The source of biomass for anaerobic digestion and the used fungal strains play important roles, mainly affecting the pollutant removal capacity, plant growth, and soil microbial community.

Further studies should be carried out regarding the changes in structure and composition of mycorrhiza, as well as soil nutrient accessibility and mycoremediation potential.

## 2. Materials and Methods

In order to enhance the reliability, transparency, and integrity of this study, a systematic review was performed according to the PRISMA Checklist 2020 (see Appendix A). Few studies were found regarding the use of digestates inoculated with fungi for contaminant removal. The objective of this work was to determine the effects of digestate application in soils, including the effect of the biomass source on the obtained digestate, the used fungal strains, the capacities for organic matter removal and plant growth improvement, and the microbial community and fungal colonization patterns.

To achieve a high-accuracy systematic review, avoid duplicates, ensure transparency, and promote academic integrity, this work was registered in the “Open Science Framework” under the project ng4kh (https://osf.io/ng4kh/, accessed on 8 September 2025), as well as in “Rayyan” (https://www.rayyan.ai/, accessed on 8 September 2025) with the number 1456753.

### 2.1. Initial Search

An initial search was carried out for English-language scientific articles published between 2015 and 2025 in the SCOPUS, SCIENCE DIRECT, PUBMED, and GOOGLE SCHOLAR databases using the Boolean operators AND and OR, as appropriate. This search was performed only by the first author of this paper in May 2025 and resulted in a substantial number of articles, of which many were found to be duplicates or not relevant.

### 2.2. Selection of Articles

Different combinations of terms were used with the Boolean connectors, corresponding to each used platform, as shown in Table 5. In total, 119 articles were initially retrieved, following which the inclusion and exclusion criteria were applied in order to include only articles which can substantially contribute to the research hypothesis. Abstracts were read by three of the authors of this paper and, in the case that insufficient information was gathered to decide whether an article should be included or not, an overview of the full text of the article was performed. To avoid risk of bias, this process was performed sequentially, not simultaneously. Furthermore, duplicates were excluded using the web-based AI application Rayyan.

#### 2.2.1. Inclusion Criteria

Only empirical research published within the period from 2015 to 2025 was considered, and no reviews, single case studies, books, or manuals were taken into consideration. In addition, it was important that the results and conclusions of the studies were clear, in order to avoid any confusion.

#### 2.2.2. Exclusion Criteria

Articles were excluded when the main topic was not related to anaerobic digestates and their use for fungal growth, or to the use of anaerobic digestates as soil amendments for plant growth or soil remediation. While some studies seemed to meet the inclusion criteria, after a detailed analysis they were excluded due to addressing the remediation process by means of bacteria, biochar, or the sole use of plants or fungi.

### 2.3. Systematic Search

Duplicates generated due to the use of different databases were excluded, with 5 duplicates found in the 119 retrieved articles. Thus, a total of 114 articles were screened, of which 54 articles were excluded due to the publication type (at this point, only empirical research was considered), leaving only 60 papers. Of these 60 papers, 33 of them were excluded as they did not contain information on fungi or digestates, instead describing the use of other microorganisms, bacteria, or biochar for soil remediation. Of the remaining 27 papers, 18 were excluded as they considered soil remediation either only with fungi or only with digestate, while not considering the simultaneous application of both. Finally, only 9 research articles (shown in Table 6) remained, including 4 articles regarding fungi growth in anaerobic digestate and 5 papers on the use of digestate inoculated with fungi for its further application in soils, for plant growth or soil remediation. Figure 4 shows the PRISMA flow diagram, summarizing every stage of the article selection process.

In order to synthesize the selected articles, different Excel tables were created, specifying the author, year, country, scale, aims, pollutant measured, biomass source for digestate, fungal strain, plant, and results obtained for each selected article. The information was separated in four different tables, explaining (a) the fungi investigated in different countries for fungi cultivation, (b) the fungi investigated in different countries as soil amendment with anaerobic digestate, (c) articles with different biomass sources for anaerobic digestate, and (d) articles with fungi strains, plants tested, and effects on plant growth and pollutant/nutrient behaviors.

Although many scientific studies regarding the use of anaerobic digestates for soil amendment or mycorrhizal approaches to pollutant removal have been published, the use of anaerobic digestate for fungal growth—both for improvement of plant development and contaminant removal—has not been widely researched. Few existing studies have demonstrated that digestate inoculated with mycorrhizae can be used as a soil amendment and for soil remediation. When considering the publication years of the reviewed articles, only two articles were published between 2015 and 2020, whereas seven articles regarding the topic were found in the last five years. These facts represent the necessity to find new methods for soil remediation.

The results of the systematic search are presented below, beginning with mycoremediation, the countries contributing to this research regarding the use of fungi for soil remediation, and the most commonly used techniques.

## 3. Results

Of the 119 articles retrieved in the first step of the systematic search, only 9 articles (representing 8%) were found to deal with the inoculation or cultivation of fungi in anaerobic digestates, with the intent of understanding their symbiotic effects on plant growth and/or metal remediation. In order to better understand the influence of anaerobic digestates on fungi and avoid risk of bias due to considering only pollutant remediation, articles regarding two main topics were included. On one hand, five articles regarding the inoculation of fungi in anaerobic digestates were considered in order to evaluate plant growth and pollutant removal. On the other hand, four articles referring to the cultivation of fungi in anaerobic digestates and its influence on contaminant removal were included.

Biogas digestates are rich in organic P and N nutrients, which contribute to plant growth and AMF colonization. Nevertheless, some authors have suggested that using anaerobic digestate could affect the AMF species richness and diversity [9]. The effects of anaerobic digestates as soil amendments on AMF communities have not been widely studied. In this systematic review, investigations regarding this topic are analyzed, especially focusing on the microbial composition, soil nutrient availability, plant growth, and pollutant removal. In order to gain a deeper understanding of the mechanisms of fungi colonization in anaerobic digestates, research articles regarding the cultivation of fungi in anaerobic digestate were included in the systematic review.

Table 7 and Table 8 provide summaries of the scientific literature considered in the systematic review. Table 7 focuses on the articles regarding the cultivation of fungi in anaerobic digestates, while Table 8 covers those on the use of digestates as soil amendments for plant growth and pollutant removal.

The effect of the biomass source from which the digestate is derived, the fungal strains, organic matter removal, and plant growth, as well as the microbial communities formed and fungal colonization are summarized in the following subsections. These results confirm the necessity of further studying the topic in order to achieve the successful remediation of polluted sites. Deeper analyses of the pollutants to be removed, the available fungal strains, and the digestate conditions need to be performed with respect to the specific required cases.

### 3.1. Biomass Source for Anaerobic Digestion

Of the articles considered, only four articles used digestates derived from the sole anaerobic digestion of cattle manure [29,47,50,53]. Table 9 shows the results regarding the different biomass sources used in the digestates. In the studies of Jasinska et al. [36] and Nesse et al. [49], digestates derived from co-digesting cattle manure and food waste were used. In the studies of O’Brien et al. [51], the growth of fungi in anaerobic digestates derived from only manure and from manure co-digested with food wastes was compared. The latter resulted in higher nutrient availability, with higher contents of K, Ca, Fe, Mg, Mn, and Na; nevertheless, mycelium colonization and fungi yield were lower, due to the relatively higher salt content of food waste and, thus, higher conductivity. On the other hand, the TN, B, Cu, and Zn contents were higher in digestate from only manure.

O’Brien et al. [51] reported that the C/N/P ratios varied according to the digestate concentration when cultivating fungi in anaerobic digestate. In particular, C/N ratios of 44:1 to 55:1 resulted in the most effective growth yield. The most beneficial N/P ratios were reported in the range of 4:1 to 12:1, where the N/P ratio changed when increasing the manure digestate concentration.

In general, the use of manure digestates as soil amendments resulted in increased microbial biomass, mycorrhizal colonization, bacterial strains, and enzymatic activity, as well as the immobilization or decreased concentrations of pollutants. In addition, improved plant growth was reported [7,8,9,29,36,45,46].

### 3.2. Researched Fungal Strains

Primarily, *A. bisporus* and *A. subrufescens* were studied for their antibiotic, anticonvulsant, and Per- and Polyfluoroalkyl Substances (PFAS) removal abilities [49,50]. Jasinska et al. [36] identified different fungal strains and observed higher degradation abilities in *Pleurotus ostreatus*, *Lentinula edodes*, and *Pleurotus eryngii*. In the articles considered for this systematic review, AMFs such as *Funneliformis mosseae*, *Rhizophagus irregularis*, or *Claroideoglomus etunicatum* were successfully inoculated in anaerobic digestates [29,48,52]. Table 10 lists the articles selected for the systematic review, describing the fungal strains, plants tested, and the reported effects on plant/fungi growth and pollutant/nutrient behaviors.

The authors Jasinska et al. [36] and Nesse et al. [50] reported that *A. bisporus* resulted in a higher biomass yield when compared to *A. subrufescens*, as well as a higher contaminant removal. In Cai et al. [29], it was concluded that inoculation with the AMF *Claroideoglomus etunicatum* resulted in a 36% increase in total nitrogen, a 60% increase in nitrate nitrogen, and a 20% increase in ammonia nitrogen. In another study, inoculation with *Funneliformis mosseae* resulted in 20%, 30%, and 25% decreases in Pb, Cd, and Zn, respectively [48]. According to Paulo et al. [52], inoculation with *Rhizophagus irregularis* led to gradual decrements of 0.77% for Cd and 0.13% for Zn. A decrease of 30%, 75%, and 100% in the phosphate, sulfate, and nitrate contents, correspondingly, were observed in [53].

### 3.3. Effects on Pollutants/Nutrients

In the selected articles, heavy metals were not the only considered pollution source. The idea was to obtain a wider coverage of the mechanisms underlying the effects of fungi and anaerobic digestates on plant growth and contaminant removal. Table 10 shows the results of the reviewed articles, describing the fungal strains, effects on plant/fungi growth, and effects on pollutants/nutrients. In the selected articles, chemical elements, drugs (i.e., antibiotics and anticonvulsants), PFAS, and other chemical compounds (e.g., nitrate, sulfate, or phosphate) were analyzed.

#### 3.3.1. Removal of Polyfluoroalkyl Substances (PFAS) and Antibiotics

The material remaining after harvesting fungi has been used to degrade polycyclic aromatic hydrocarbons (PAHs) and other persistent organic pollutants, as well as for heavy metal biosorption or even as a biological control agent for nematode pests in soils [45,54]. Nesse et al. [50] assessed the uptake of per- and polyfluoroalkyl substances (PFAS) during fungi cultivation, resulting in a lower accumulation of PFAS in fungi through the use of digestate. They found that *A. bisporus* revealed a higher PFAS concentration, when compared to *A. subrufescens*. Ultrashort-chain PFAS showed much greater accumulation compared to long-chain PFAS. Additionally, temperature was found to have a direct influence on the degradation rate of organic matter, as the digestate composition may vary accordingly, influencing the sorption and bioavailability of pollutants. Meanwhile, short-chained acids are less affected by pH changes than long-chained ones.

The capacity of fungi to remove two antibiotics and one anticonvulsant was tested in [49], with the results indicating that antibiotics were not accumulated in fungi. The biodegradation of different chemical elements was also analyzed, including Ca, Co, Cr, Fe, K, Mg, Mn, Mo, Na, Ni, Se, Si, and Zn. PFAS are used in many industrial and consumer products and persist in the environment, eventually accumulating in the human body. In Nesse et al. [49], the removal of antibiotics by two different fungi, *A. subrufescens* and *A. bisporus*, was compared. In general, the ciprofloxacin removal rate was 83–90% and that for carbamazepine was 57–97%, even though the antibiotic content was low. *A. subrufescens* showed higher efficiency in terms of contaminant removal. This fungus is considered tropical and was cultivated at higher temperatures (25 °C for A. *subrufescens* and 17–18 °C for *A. bisporus)*.

#### 3.3.2. Pollutants/Nutrients

Table 10 lists the pollutants/nutrients analyzed in the selected research articles. Jasinska et al. [36] reported Zn as antagonist of some metals, such as Cd, Pb, and Ni, with the presence of Zn in fungi hindering high concentrations of other toxic metals. Cd accumulation was found to be mycelium age-specific: the older the mycelium, the less Cd accumulated.

The phosphate and sulfate contents were compared using inorganic fertilizer and a digestate as organic fertilizers [53]. It was reported that the contents of both micronutrients were higher under the anaerobic digestate treatment. Nitrate content was lower with the digestate, in comparison to inorganic fertilizer, whereas the uptake of P, S, N, K, Ca, Zn, B, and Mg was higher. The organic fertilizer resulted in higher pH and available P concentration, but a lower total grass dry matter. Bacteria utilizing sulfonate and phosphonate were up to five times more abundant under the organic fertilizer treatment. The abundance of phytate- and calcium phosphate-utilizing bacteria did not significantly differ between the treatments.

In Fontaine et al. [48], a significant decrease in Pb was reported when inoculating coriander with AMF using an anaerobic digestate as a soil amendment, resulting in higher Pb extractability. The concentrations of trace elements did not change in the soil and vegetation. Higher bioavailability of Cd, Pb, and Zn was observed when no amendment was applied to the soil. When the vegetated soil was inoculated with AMF, the bioavailability of these elements decreased; meanwhile, when non-amended soil was inoculated with AMF, Cd, Zn, and Cu bioavailability increased. Cd was the predominantly accumulated element in the coriander shoots. Fontaine et al. [48] demonstrated decreases in Cd and Zn extractability (i.e., bioavailability), as well as decreased Pb and Cd accumulation in coriander shoots.

In the experiments conducted by O’Brien et al. [51], the N/P and C/N ratios decreased with increasing digestate addition, demonstrating the P fertilization effect of adding digestate. The C/N ratio was positively correlated with the biomass yield. The most effective C/N ratios ranged between 44:1 and 55:1. Other authors have reported substrate combinations, where a C/N ratio from 72:1 to 81:1 produced the highest yields of the fungus *P. ostreatus* [55]. The most productive N/P ranged between 4:1 and 12:1.

### 3.4. Effects on Plant Growth

Anaerobic digestate can be inoculated with fungi for its further use as an organic amendment, allowing for further soil quality and health improvements. Table 9 summarizes the plant growth outcomes reported in the articles selected for this systematic review.

Most of the articles selected for this review showed a significant increase in plant growth when applying high digestate concentrations in the soil, especially when the biomass source of the digestate was manure [29,48,51,52]. In contrast, Ikoyi et al. [53] reported a lower plant growth when applying digestate. In Fontaine et al. [48], it was reported that using anaerobic sludge in soil inoculated with AMF *Funneliformis* mosseae resulted in better growth of coriander plants as well as Cd and Zn immobilization in soil and, thus, lower Cd plant uptake. The used anaerobic digestate could improve the physical characteristics of soils, such as air permeability, water retention, stability, aggregation, and resistance to soil erosion, allowing nutrients to be released slowly to support long-term plant growth.

### 3.5. Effects on Fungal Population

Several authors have demonstrated that the application of soil amendments in the form of anaerobic digestates results in the enrichment of microbial communities in soil [47,48,53]. Yu et al. [47] documented significant increases in the AMFs *Glomus* and *Paraglomus* when using anaerobic digestate, which increased soil fertility and activated a wider range of soil microbiota; in particular, important increases in the Shannon diversity and species richness of AMFs were distinguished. When applying anaerobic digestate in soil, more organic matter needs to be decomposed and, thus, a wider variety of microorganisms is needed. It has also been demonstrated that different concentrations of anaerobic digestate promote the growth of different fungi. *Glomus* showed a higher abundance when applying increased concentrations or anaerobic digestate, while the growth of *Paraglomus* was higher at lower anaerobic digestate concentrations. The yield of *Pennisetum* doubled when planted with the highest digestate concentration tested. AMF inoculation did not lead to any significant difference. Thus, the effects of AMF inoculation were mainly manifested in soil properties and microbial diversity, instead of plant growth.

Paulo et al. [53] reported significantly higher bacteria feeders, nematode abundance, and root colonization in mycorrhizal arbuscules, hyphae, and vesicles when using anaerobic digestate as a fertilizer compared to the use of inorganic fertilizer. Analysis of nematode abundance and bacteria feeders revealed a higher prevalence of bacteria-feeding nematodes from the families *Cephalobidae* and *Rhabditidae*. Paulo et al. [53] found 30% more fungal arbuscles and hyphae, as well as 20% more vesicles, when applying digestate to *Lolium perenne* (ryegrass) inoculated with AMF.

Fontaine et al. [48] reported a significant increase in the total microbial biomass when using anaerobic digestate as a soil amendment. Although the soil metabolic potential and functional richness were lower in non-amended soils, the values increased when growing coriander in these soils. Dehydrogenase activity was not altered in non-amended soils. *Phosphomonoesterase* activity was not affected by either amendment or growing coriander.

After AMF addition, Cai et al. [29] reported an improvement in the richness and diversity of fungi in hybrid *Pennisetum* soil; however, the effect of AMF addition was reduced after application of an anaerobic digestate.

## 4. Discussion

Although many studies on the sole use of anaerobic digestates as soil amendments or the sole use of mycoremediation for soil pollutant removal have been published, the simultaneous use of both technologies has not been widely researched. Such approaches show promise as a relevant strategy for promoting the circular bio-economy, sustainable agriculture, and environmental biotechnology.

The use of digestate generally increases microbial richness and enzymatic activity, leading to improved soil stabilization and fertility. The decomposition of organic matter derived from digestate requires a broader spectrum of microbial taxa, which promotes bacterial proliferation and improves mycorrhizal colonization. However, low digestate concentrations were associated with lower AMF diversity, while higher concentrations favored the dominance of genera such as *Glomus* and *Paraglomus*. These results suggest that digestate concentration is an important determinant of microbial community composition. Table 11 lists the effect of digestate concentration on fungal diversity and colonization.

According to Yu et al. [47], the lower concentration of anaerobic digestate resulted in a decrease in AMF diversity, while a higher digestate concentration led to significant increases in the abundance of the dominant genera *Glomus* and *Paraglomus*. Furthermore, soil fertility and the presence of soil microorganisms increased with the application of digestate, due to the fact that organic fertilizers provide a higher amount of organic matter to be decomposed. Cai et al. [29] observed a higher organic matter removal when applying the highest digestate concentration, with a simultaneous increase in soil nutrient content. It can be assumed that, in this scenario, microorganisms have sufficient nutrient sources to maintain a higher reproduction rate, leading to higher organic matter decomposition.

It has been demonstrated that the application of organic amendments enriches microbial communities in soils, whereas plant-associated microorganisms may reduce the uptake of TEs by plants. This could be a reason why reported significant decreases in extractable Cd and Zn, while Pb and Cd were accumulated in coriander shoots. Furthermore, rhizosphere microorganisms have the capacity to regulate plant nutrient uptake and TE bioavailability through mechanisms such as oxidation, reduction, complexation, immobilization, adsorption, or dissolution [28]. The reviewed results suggest that anaerobic digestates have greater impacts on the diversity of fungal communities in rhizospheric soil, compared to bacterial communities.

AMFs play a crucial role in modulating nutrient availability by supplying P and other nutrients to plants in exchange for carbohydrates, while also altering bacterial community structure. AMF hyphae have been shown to capture bacterial strains that would otherwise limit nutrient uptake, while secreting enzymes that break down organic matter into simpler compounds. Moreover, AMFs contribute to TE immobilization via glomalin production, adsorption onto hyphae, and sequestration in fungal structures such as vacuoles.

Fungal strains differ in their degradative ability and nutrient accumulation when grown in digestate. *Pleurotus ostreatus, Lentinula edodes*, and *Pleurotus eryngii* exhibited high growth rates and efficient degradation, while *Agaricus bisporus* and *Agaricus subrufescens* showed contrasting abilities in pollutant absorption and growth performance. For instance, *A. subrufescens* demonstrated reduced uptake of PFAS compared to *A. bisporus*, but achieved greater growth, suggesting a trade-off between pollutant absorption and fungal biomass production. Table 12 summarizes the results of the selected articles regarding fungal strains and their tolerance to anaerobic digestate.

Jasinska et al. [36] reported a higher degradative ability and growth rate in anaerobic digestate for *Pleurotus ostreatus*, *Lentinula edodes,* and *Pleurotus eryngii*. This difference depends on both the fungal species and substrate characteristics, such as its C/N ratio, pH, and contaminants. These authors reported that the concentrations of trace elements in the anaerobic digestate directly affect the concentrations of trace elements in fungi.

Although anaerobic digestates represent adequate nutrient sources, nutrient availability depends greatly on the fungal species. *Agaricus bisporus* provides significant amounts of K and Na, while *Agaricus subrufescens* is an important source of Cu and Zn, indicating the specific capacity of fungi to bioaccumulate certain elements. Jasinska et al. [36] suggested that Si content could augment microbial interactions, whereas plant colonization by pathogens could be hindered.

From the selected articles, it was observed that *A. subrufescens* showed lower growth when higher decreases in pollutants such as Ca, Fe, Na, Ni, Cr, Si, Mg, and Zn were detected. *A. subrufescens* showed a lower PFAS uptake (by half) than *A. bisporus*, whereas *A. subrufescens* grew 50% more than *A. bisporus*. These results suggest that higher pollutant absorption may hinder fungal growth. In contrast, Nesse et al. [49] compared the antibiotic and anticonvulsant removal abilities of both fungi. *A. subrufescens* showed a higher efficiency in contaminant removal than *A. bisporus*, whereas the contaminant concentrations in this fungus were lower at the end of the experiments. However, no fungal growth values were reported.

The removal of pharmaceuticals also varied among fungal species. *A. subrufescens* exhibited a greater removal efficiency of antibiotics and anticonvulsants than *A. bisporus,* although showed a lower intracellular accumulation, indicating extracellular degradation or strong sorption mechanisms. Antibiotics, especially fluoroquinolones, were more easily degraded due to photolysis and sorption, while carbamazepine showed low removal due to its strong sorption properties and recalcitrance [49]. Furthermore, non-extractable residues could have formed, contributing to the observed decline in the antibiotic concentration. It has been reported that fluoroquinolones have a strong affinity to sorb to solids, being electrostatically bound to the substrate. Sulfonamides, such as the other antibiotic tested, possess an amine electron-donating group, making them more susceptible to degradation. Carbamazepine showed stronger sorption than the antibiotics. Table 13 shows the contaminants and fungal mechanisms of removal/immobilization found in the selected articles.

Fontaine et al. [48] reported that the application of anaerobic digestate in coriander cultivation under AMF inoculation plays a crucial role in restoring soil functionality. Soil processes, such as the formation and decomposition of organic matter, as well as respiration and nutrition cycles, are impacted by microbial communities. Furthermore, the composition and structure of soil can be regulated by AMFs.

AMFs provide P and other nutrients to plants and receive carbohydrates in exchange, while protecting them from drought and pathogens [47]. AMFs have been reported to change the bacterial community composition; furthermore, AMF hyphae have been shown to capture different bacterial strains which otherwise would affect plant nutrient uptake and growth. Jasinska et al. [36] reported that, during colonization, the fungal mycelium decomposes the growing substrate by secreting enzymes that have the capacity to break down organic matter into simpler compounds. Furthermore, fungi can accumulate trace elements or even dissolve metals through root exudates.

The interaction between aromatic plants (e.g., coriander), AMF and digestate makes the complexity of TE dynamics even clearer. Plant exudates containing organic acids alter soil chemistry and form organometallic complexes that reduce the bioavailable fractions of Cd, Zn, and Pb. Digestate and AMF together enhance TE mobilization, while the bioavailability of Pb increase in soils without digestate. These results emphasize the importance of soil pH buffering, organic matter input and microbial interactions in shaping TE dynamics. Several studies have shown that digestates increase the immobilization of metals and metalloids, as a result of different processes; for example, adsorption onto mineral surfaces, formation of stable compounds and organic ligands, surface precipitation, or even ion exchange. These processes have only been suggested, and not deeply researched, in previous studies. Palansooriya et al. [56] suggested that applying composted anaerobic digestate could minimize the mobility of TEs and thus toxicity, as a result of the increased dissolved organic matter. The study of Fontaine et al. [48] reported increased total and organic C, P, and Mg contents when applying composted anaerobic digestate, improving soil TE immobilization and plant nutrition, ultimately resulting in enhanced coriander shoot growth. However, the same authors reported no significant effect on shoot growth when inoculating with AMF. Mechanisms related to TE immobilization through AMF inoculation include the production of glomalin-related soil proteins in the mycorrhizosphere, TE accumulation in fungal structures (e.g., vacuoles or fungal vesicles in mycorrhizal roots), and adsorption of TEs by extraradical hyphae [57]. Due to the high P and N contents in anaerobic digestates, increased available P in soil has been reported [47].

Nutrient availability is a key factor for efficient plant and microbial community growth. The use of anaerobic digestates could guarantee the availability of essential micronutrients such as N, P, S, the C/N ratio, or even certain heavy metals (i.e., Cd, Cr, Cu, Fe, P, or Zn). Authors have confirmed that inoculation with AMF resulted in increases in essential nutrients, such as TN or available P [29,47,53]. Decreases in the phosphate, sulfate, and nitrate contents were observed in the experiments of [53], with associated increases in the abundance of nematodes, bacterial strains, mycorrhizal colonization, and enzymatic activity. It can be speculated that the absorbed dietary phosphorous (in the form of phosphate) was used for mycorrhizal colonization and the growth of bacterial strains or nematodes, instead of remaining as dissolved ionic phosphate in the soil.

In Fontaine et al. [48], it was reported that Pb bioavailability increased in coriander under the no-digestate treatment, whereas Pb bioavailability decreased with AMF and digestate. Pb bioavailability increases when its dissolved ionic form increases, or when its transport across membranes is facilitated. These scenarios can take place at a low soil pH, or in the presence of specific bacteria. Meanwhile, low Pb bioavailability occurs at increased pH levels or in the presence of organic matter. Materials such as compost or digestates immobilize lead through binding to their organic and mineral components, thus making it less bioavailable [58].

Fungi show advantages over plant production due to the fact that they can be grown in processed substrates, such as composted, pasteurized, or even sterilized media. In the process of fungal growth, some toxic compounds in the substrate could be decomposed. Jasinska et al. [36] reported that fungal mycelium grows within the substrate, decomposing the material, whereas colonization takes place through secreted enzymes breaking down complex organic matter into simpler components. Senila et al. [59] demonstrated that fungi root exudates can dissolve metals from the substrate in the mycelial zone, influencing mineral adsorption.

Furthermore, the biomass source utilized during the process of anaerobic digestion is a key factor affecting nutrient availability. It has been confirmed, by several authors that the sole use of manure in anaerobic digestion resulted in higher plant yield and fungal colonization associated with the derived digestate. In O’Brien et al. [51], it was reported that when co-digesting manure with food waste, fungal colonization was hindered and, thus, plant yields were inhibited. This could have occurred for several reasons. The composition of organic wastes may vary significantly depending on their source, reflecting specific physical and chemical characteristics [60]. A high conductivity could have been generated due to the presence of organic wastes, affecting fungal growth. In Arifan et al. [61], inhibition was reported when co-digesting manure with organic wastes: the greater the amount of cabbage waste, the lower the efficacy of the anaerobic digestate. This reflects the idea that organic wastes could include toxic substances or inhibitors, which are not perceived at a glance but ultimately inhibit plant growth or even soil stabilization. Another aspect to consider is osmotic stress, which affects the C/N ratio in plants. Throughout their entire life cycle, microorganisms are continuously exposed to abiotic stress factors such as temperature fluctuations, salinity, oxidation, or osmotic stress [62], the latter of which is caused by changes in solute concentrations inside cells as a result of drought or salinity stress [63]. The C/N is affected when the plant accumulates osmotic solutions to balance water loss, thus altering photosynthesis and nitrogen metabolism in the plant [64].

It has been reported that the application of highly concentrated anaerobic digestate resulted in significant enhancement of the species richness and Shannon diversity of AMFs in the rhizosphere of a poplar plantation. Increased microbial diversity can occur when soil properties change, such as increased P availability and decreased C/N ratio [47]. This results in better N availability, promoting plant growth. Furthermore, the C/N ratio is more important than the C concentration [65]. In fact, Ikoyi et al. [53] observed N limitation when the C/N ratio was higher than 20, leading to decreased plant growth: a high C/N ratio is associated with a low N content. As N is an important component of chlorophyll, high amounts of N are required for plant growth. Thus, studies reporting significant plant growth also reported that the N level remained the same or even increased. Furthermore, according to Gao et al. [65], N has a higher impact than C on the growth of species of the fungal genus *Entomophthorales*. It has also been reported that higher C:N ratios reduced sporulation, whereas C/N ratios in the range of 10:1 to 40:1 were reported to result in higher yields, compared to those from 80:1 to 160:1. In the analysis of O’Brien et al. [51], C/N ratios of 44:1 to 55:1 resulted in the most effective growth yield, while the optimal N/P ratios were in the range of 4:1 to 12:1.

Fungal C/N/P stoichiometric requirements have not been widely researched, although they play a very important role in biogeochemical cycling. Some fungi—e.g., mycorrhizae—use organic forms of nitrogen and phosphorus which otherwise would not be available to plant roots. Variations are associated with different functional guilds within fungal groups. C/N has been reported to be lower and N/P higher in EcM fungi, compared to saprotrophic fungi, which can be attributed to EcM fungi having evolved over a longer time, in comparison to saprotrophic fungi [66,67].

The C/N and C/P ratio of digestate, which is largely influenced by the type of source material (slurry or co-fermented organic waste), has a strong impact on microbial colonization and plant growth. Digestates derived from slurry, which are characterized by a lower C/N ratio, consistently promoted greater fungal colonization and higher plant yield compared to digestates from mixed organic waste. High C/N ratios led to nitrogen limitation, reduced fungal spore formation and reduced plant performance. The conductivity and salt content of the digestate, which depend on the composition of the feedstock, also led to osmotic stress, which inhibited microbial colonization. Table 14 outline soil and environmental factors affecting fungal efficiency, according to the papers selected.

In O’Brien et al. [51], it was reported that digestate derived from manure and organic wastes showed a lower colonization rate than digestate from only manure. When the manure content increased, the C/N and C/P ratios decreased from 29/1 to 22/1 and from 186/1 to 127/1, correspondingly, while the N/P ratio remained same. Manure is rich in N and shows a low C/N ratio, between 7/1 and 25/1. Food waste typically has a higher C/N, of 20/1 up to 98/1 [68]. A very high C/N leads to a lack of nitrogen for microbial growth, as was the case in O’Brien et al. [51]. Organic wastes include agricultural wastes, market wastes, kitchen wastes, or even urban wastes. As such, the composition of organic wastes may vary significantly, and may provide beneficial amounts of nutrients (e.g., K, N, or P) for the growth of fungi and plants; however, these wastes may also have high loads of heavy metals, pharmaceutical products, or other toxic compounds [69]. In O’Brien et al. [51], it was reported that conductivity—and, thus, salinity—is an important issue leading to lower colonization rates. This may have also been the reason for the low plant yields reported in O’Brien et al. [51] when co-digesting with manure.

Regarding pH and temperature, it has been suggested that, for optimal plant and/or fungi growth, a balanced C/N ratio and a suitable pH must be guaranteed. In Nesse et al. [50], *A. subrufescens* cultivated at 20 °C and neutral pH showed higher contaminant removal than *A. bisporus* grown at 18 °C and tolerating slightly alkaline conditions. Digestate and fungal activity together buffered the pH value of the soil in the neutral range, promoting the decomposition of organic matter and improving the availability of nutrients. These synergistic effects are of central importance for soil fertility, TE immobilization, and sustainable remediation processes.

The authors of Nesse et al. [50] reported that, when inoculating with fungi, organic acids in soil were formed through organic matter degradation, leading to a decrease in pH and an increase in dissolved organic carbon. Thus, organic acids resulted in higher organic matter degradation. In comparison to tests with digestate, the control test with no fungal cultivation showed lower organic matter degradation and increased pH. The content of organic matter in soil has a direct impact on soil functions and the global carbon cycle. After decomposition, it determines the nutrient cycle and can take up C from the atmosphere through photosynthesis, enabling its long-term sequestration in soils. Moreover, organic matter enhances the soil’s fertility, water-holding capacity, and nutrient cycling, supporting sustainable agriculture and ecosystem health [70]. Soil management practices affect the organic matter stock, and an appropriate soil management approach can even enhance nutrient availability and C sequestration. Physical and chemical fractionation methods should be applied to differentiate the functional soil organic carbon fractions, thus optimizing carbon sequestration potential, guaranteeing the preservation of soil organic matter, and ultimately improving soil fertility in the long-term [71].

In the experiments reported in Fontaine et al. [48], the pH value did not change significantly when applying digestate to soil, maintaining pH in the neutral range. Nevertheless, the available C, Mg, and P contents increased, resulting not only in increased trace element immobilization but also an improved nutritional state in plants. Therefore, pH was buffered and efficient plant growth was achieved.

Finally, how digestates modulate fungal colonization and diversity must be discussed. The application of organic amendments has been demonstrated to enrich microbial communities, as a higher amount of organic matter must be decomposed. It has been confirmed by several authors that a low digestate concentration leads to diminished AMF diversity [47,48]. Furthermore, digestates increase the immobilization of metals and metalloids, as a result of processes such as adsorption onto mineral surfaces, the formation of stable compounds and organic ligands, surface precipitation, or even ion exchange [48]. The application of digestates guarantees the availability of essential micronutrients such as N, P, S, or C, which enable the reproduction of microorganisms, such as AMFs, which are capable of decomposing organic matter or even removing toxic compounds [47]. The diversity of AMFs and the microbial community plays a crucial role in restoring soil functionality, providing nutrients such as P and N while receiving carbohydrates in exchange. Soil nutrient availability is facilitated by healthy soil microbiota (i.e., archaea, bacteria, and fungi), which mineralize, solubilize, and facilitate the uptake of nutrients. The carbon content in digestates has significant influences on soil heterotrophic microbial dynamics and nutrient availability. If the C content in a digestate is low, it will be reflected in the microbial community, favoring fast-growing bacteria; meanwhile, if the C content of the digestate is increased, soil microbial abundance and diversity also increase, thus favoring slower-growing fungi and Gram-positive bacteria [72].

## 5. Conclusions

Recent studies emphasize the complementary role of anaerobic digestates and AMF in restoring soil functionality, improving nutrient cycling and reducing the bioavailability of TE. While the use of digestate or AMFs alone has been widely documented, their combined application is still under-researched despite their potential to promote sustainable agriculture and circular bio economy strategies.

For this systematic review, the effects of anaerobic digestates as organic soil amendments inoculated with fungi were analyzed. To obtain results with better reliability and transparency, this systematic review was performed according to the PRISMA guidelines, with the systematic selection process resulting in the final retrieval of nine scientific articles published within the last ten years. The selected articles described scientific research regarding the use of anaerobic digestates inoculated with fungi, considering their effects on plant growth and/or pollutant removal. Knowledge regarding this topic remains scarce. The aim of this work was to determine the effects of digestate application on soils, including the effect of the biomass source from which the digestate was obtained, the fungal strains, and plant growth, as well as the microbial communities formed and fungal colonization.

The findings on the use of anaerobic digestates as soil amendments inoculated with fungi underline their long-term ecological potential for the remediation of polluted soils. This strategy is proving to be a promising and cost-effective tool for improving soil functionality and stability, rather than directly promoting plant growth.

Through the application of digestates as soil amendments, the microbial population is enriched, thus increasing metal uptake through mechanisms of soil remediation including oxidation, reduction, complexion, immobilization, adsorption, or dissolution. The change in the microbial population is reflected in the availability of P and the C/N ratio. If the nitrogen content is limited, thus leading to a high C/N ratio, plant growth will be affected as N is an important component of chlorophyll—a green pigment which is essential for plant growth.

The origin of the biomass used for anaerobic digestion plays a crucial role in determining the quality of the digestate. If the digestate is derived exclusively from cattle manure, its application to fungi-inoculated soils will result in increased organic matter decomposition and mycelial colonization, as well as lower salinity and electrical conductivity. In contrast, the addition of mixed organic waste can increase the concentrations of TEs and HMs, which can impair the development of the microbial community and limit the efficiency of remediation mechanisms. Overall, the use of anaerobic digestates, especially those derived from cattle manure, promotes microbial diversity and improves the processes associated with soil remediation and stabilization. The benefits observed in terms of plant growth are less consistent and appear to be secondary to the effects on soil health.

While the fungal strain selected for inoculation is a key factor, it must be evaluated in conjunction with other variables such as digestate composition, application concentration, pH, temperature, and contaminant type. The specific fungal strain may affect soil stabilization, pollutant removal, and plant growth.

Some authors have specified that pH and temperature in particular affect the degradation of organic matter and removal of contaminants. The presence of organic acids results in higher organic matter degradation and decreased pH. AMFs stimulate organic acid production, thus breaking down organic matter into simpler compounds. Digested manure is a good source of organic acids—another reason why solely using manure as a source for digestate resulted in better plant growth and contaminant degradation, as reported by several authors. It was also reported that temperature conditions around 20 °C resulted in more efficient performance, when compared to colder temperatures (around 17 °C).

The combined application of fungi and digestate contributed to a reduction in Pb content and increased bioavailability, while improving soil pH and nutrient availability. Aromatic plants such as coriander promote metal complexation and immobilization through their root exudates and the release of organic acids, especially when applied together with anaerobic digestates. The interaction of digestate addition influenced AMF diversity and nutrient availability, thus directly affecting soil fertility and plant growth.

Interestingly, to improve the bioavailability of certain heavy metals such as Cd, Zn, and Cu, the absence of vegetation during certain phases of remediation may be necessary—an aspect that warrants further investigation.

The limitations of this review include that the results and efficacy vary according to the fungal strain, plant grown, pollutant to be removed, and specific environmental conditions. A further limitation of the selected articles is the lack of results oriented to real field conditions, as no research was found in this regard. Future research should focus on a goal-oriented pollutant-specific remediation strategy using digestates derived solely from manure, ideally in combination with tropical fungal strains to maximize their efficacy under specific environmental conditions.

In summary, soil remediation is important to safeguard public health and the environment. New cost-efficient practices can be proposed for sustainable soil management, and future research should focus on the interactions with pollutants in soil amended with digestates and fungi, validating these novel methods under natural conditions.

## Figures and Tables

**Figure 1 biology-14-01579-f001:**
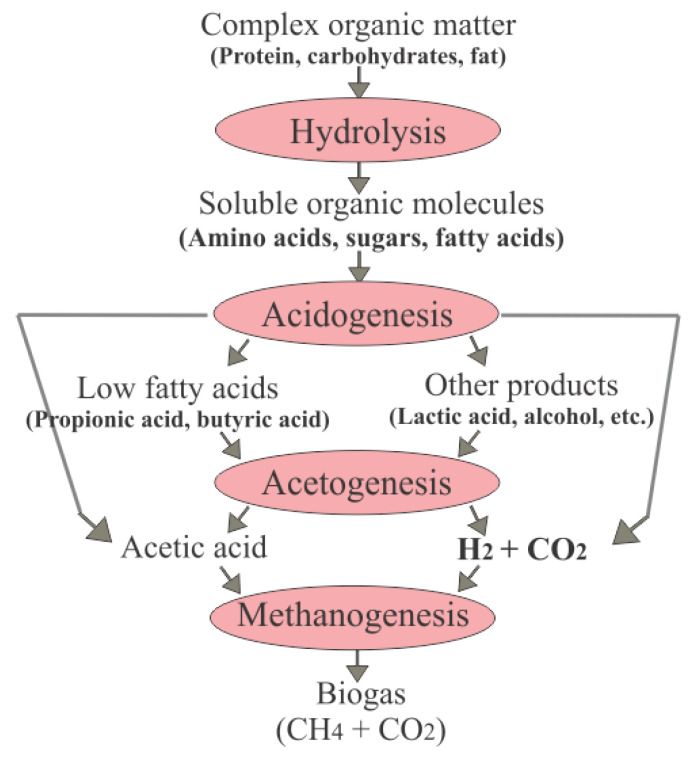
Stages in biogas production (CH_4_: methane; CO_2_: carbon dioxide; H_2_: hydrogen).

**Figure 2 biology-14-01579-f002:**
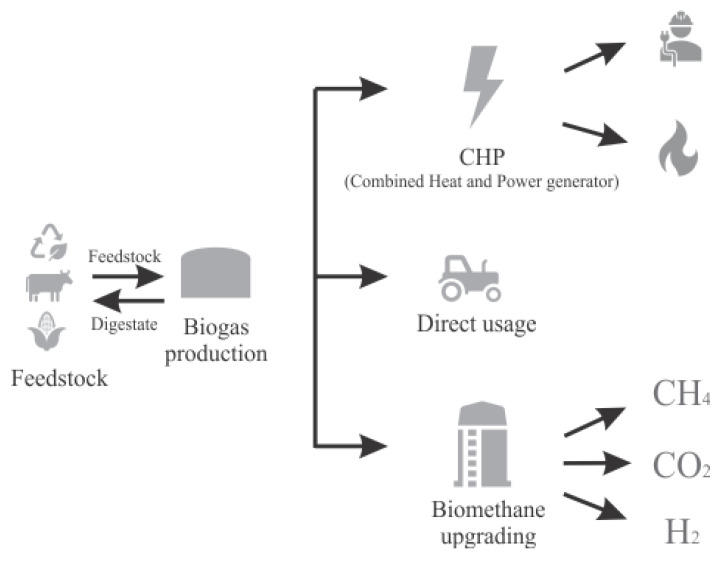
Potential utilization pathways for biogas.

**Figure 3 biology-14-01579-f003:**
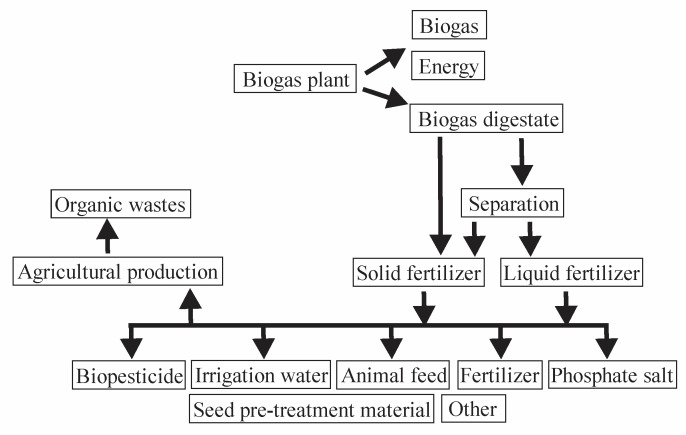
Possible uses of biogas digestate.

**Figure 4 biology-14-01579-f004:**
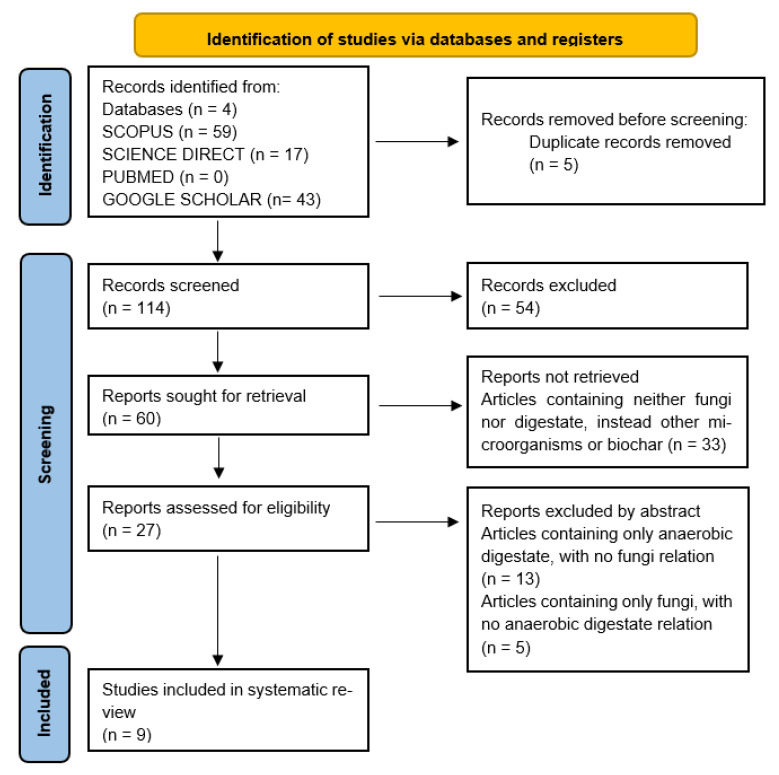
Flowchart of stages of the article selection process.

**Table 1 biology-14-01579-t001:** Doubling time of different bacteria/archaea involved in biogas production.

Bacteria/Archaea Group	Doubling Time (h)
Hydrolytic and acidogenic bacteria	
*Bacteroides*	<24
*Clostridium*	24–36
Acetogenic bacteria	
Syntrophobacter	40–60
Syntrophomonas	72–132
Methanogenic Archaea	
Methanobacterium	12–60
Methanosarcina	120–360
Methanococcus/Methanosaeta	240

**Table 2 biology-14-01579-t002:** Typical biogas composition.

Component	Concentration
Methane (CH_4_)	50–75%
Carbon dioxide (CO_2_)	25–50%
Nitrogen (N_2_)	0–10%
Hydrogen (H_2_)	0–1%
Hydrogen sulfide (H_2_S)	0–3%
Oxygen (O_2_)	<0%

**Table 3 biology-14-01579-t003:** Specific biogas and methane production levels for each substance group.

	Theoretical COD (g O_2_·g^−1^)	Biogas (mL·g^−1^ VS)	CH_4_ (%)	CH_4_ (mL·g^−1^ VS)	Biogas (mL·g^−1^ COD)	CH_4_ (mL·g^−1^ COD)
Protein	1.13	750	50	375	664	332
Fat	1.6	800	60	480	500	300
Carbohydrates	2.03	1390	72	1001	685	493

**Table 4 biology-14-01579-t004:** Limit values permitted in anaerobic digestate intended for agricultural use.

Trace Element in Anaerobic Digestate	Limit Values in mg·kg^−1^ (Dry Matter)
Cd	20–40
Cu	1000–1750
Hg	16–25
Ni	300–400
Pb	750–1200
Zn	2500–4000

**Table 5 biology-14-01579-t005:** Search strategy with Boolean operators.

Database	Combination of Terms and Boolean Operators	Number of Articles Found
SCOPUS	(biogas AND digestate AND mycorrhiza AND metal AND remediation) OR (biogas AND digestate AND mycoremediation) OR (biogas AND slurry AND mycorrhiza AND metal AND remediation) OR (biogas AND slurry AND mycoremediation) AND PUBYEAR > 2014 AND PUBYEAR < 2026 AND (LIMIT-TO (DOCTYPE, “ar”))	59
GOOGLE SCHOLAR	(anaerobic digestate + mycorrhiza + metal remediation) OR (biogas digestate + mycoremediation) OR (biogas slurry + mycorrhiza + metal + remediation) OR (biogas + slurry + mycoremediation)	43
SCIENCE DIRECT	(anaerobic AND digestate AND mycorrhiza AND metal AND remediation) OR (biogas AND digestate AND mycoremediation)	17
PUBMED	(anaerobic AND digestate AND mycorrhiza AND metal AND remediation) OR (biogas AND digestate AND mycoremediation)	0

**Table 6 biology-14-01579-t006:** Studies selected for the systematic review.

Number	Year	Title	Reference
1	2024	Effects of Arbuscular Mycorrhizal Fungi and Biogas Slurry Application on Plant Growth, Soil Composition, and Microbial Communities of Hybrid *Pennisetum*	[29]
2	2024	Mushroom production on digestate: Mineral composition of cultivation compost, mushrooms, spent mushroom compost and spent casing	[36]
3	2024	Low uptake of pharmaceuticals in edible mushrooms grown in polluted biogas digestate	[49]
4	2023	Uptake of Ultrashort Chain, Emerging, and Legacy Per- and Polyfluoroalkyl Substances (PFAS) in Edible Mushrooms (*Agaricus* spp.) Grown in a Polluted Substrate	[50]
5	2023	The Potential of Bioaugmentation-Assisted Phytoremediation Derived Maize Biomass for the Production of Biomethane via Anaerobic Digestion	[52]
6	2020	Impacts of Biogas Slurry Fertilization on Arbuscular Mycorrhizal Fungal Communities in the Rhizospheric Soil of Poplar Plantations	[47]
7	2020	Coriander (*Coriandrum sativum* L.) in Combination with Organic Amendments and Arbuscular Mycorrhizal Inoculation: An Efficient Option for the Phytomanagement of Trace Elements-Polluted Soils	[48]
8	2020	Responses of soil microbiota and nematodes to application of organic and inorganic fertilizers in grassland columns	[53]
9	2019	Integrating anaerobic co-digestion of dairy manure and food waste with cultivation of edible mushrooms for nutrient recovery	[51]

**Table 7 biology-14-01579-t007:** Fungi investigated in different countries for cultivation in digestates.

Year	Country	Fungi	Aim of the Analysis	Reference
2024	Poland	*Agaricus bisporus* and *Agaricus subrufescens*	HM accumulation	[36]
2024	Norway	*Agaricus bisporus* and *Agaricus subrufescens*	Drug accumulation	[49]
2023	Norway	*Agaricus bisporus* and *Agaricus subrufescens*	Per- and polyfluoroalkyl substances (PFAS) accumulation	[50]
2019	USA	*Pleurotus ostreatus*	HM accumulation	[51]

**Table 8 biology-14-01579-t008:** Fungi investigated in different countries as soil amendment with anaerobic digestates.

Year	Country	Fungi	Plant	Aim of the Analysis	Reference
2024	China	AMF (*Claroideoglomus etunicatum*)	*Pennisetum*	Effect of AMF inoculation in digestate for plant growth.	[29]
2023	Portugal	AMF (*Rhizophagus irregularis*)	*Zea mays*	Effect of AMF inoculation in digestate for plant growth.	[52]
2022	China	AMF (strain not specified)	Poplar	Fungal strain identification.	[47]
2022	France	AMF (*Funneliformis mosseae*)	*Coriandrum sativum* L.	Effect of AMF inoculation in digestate for plant growth.	[48]
2020	Ireland	AMF (strain not specified)	*Lolium perenne*	Effect of AMF inoculation in digestate for plant growth.	[53]

**Table 9 biology-14-01579-t009:** Articles using different biomass sources for anaerobic digestates.

Biomass Source for Anaerobic Digester	Parameters Measured	Digestate Effect	Reference
Anaerobically digested cattle manure	Organic matter Total nitrogen (TN) Nitrate nitrogen Ammonia nitrogen Available phosphorous (P)	Plant growth was doubled at highest digestate concentration tested (13,216.67 kg/hm^2^) Almost all parameters increased with increasing digestate application Only soil organic matter decreased	[29]
Anaerobically digested cattle manure	AMF community identification P and nitrate nitrogen	Highest digestate concentration showed the highest Shannon diversity 36.2–42.7% of AMF diversity was *Glomerales* *Glomus* and *Paraglomus* abundance was doubled with digestate application P and nitrate nitrogen increased with digestate	[47]
Anaerobically digested cattle manure	Per- and polyfluoroalkyl substances (PFAS)	Low accumulation of PFAS in fungi	[50]
Anaerobically digested cattle manure	Soil microbiota Available phosphorus (P) Sulfur (S)	Higher abundance of nematodes, bacterial strains, mycorrhizal colonization, enzymes, and available P	[53]
Anaerobically digested cattle manure and food waste	45 elements: Major essential elements: Ca, K, Mg, and Na Trace elements: B, Co, Cr, Cu (copper), Fe, Mn, Mo, Ni, Se, Si, and Zn Trace elements with detrimental health effect: As, Ba, Be, Cd, Hg, Pb, and Tl Nutritionally non-essential elements: Al, Bi, Ce, Ge, Ir, Li, Nd, Os, Pr, Pt, Rb, Sb, Sn, Sr, Ta, Te, Ti, U, V, W, and Zr	Decreases in Al, As, B, Ca, Cd, Cr, Fe, K, Mg, Mn, Ni, Pb, and Zn after every harvest	[36]
Anaerobically digested cattle manure and food waste	Sulfonamide antibiotic Fluoroquinolone antibiotic Anticonvulsant carbamazepine	Antibiotics were easily removed by fungi Anticonvulsant was accumulated in small amounts only by *A. bisporus*	[49]
Anaerobically digested cattle manure and food waste	Total solids, N, C, As, P, Ca, K, Mg, Al, B, Cu, Fe, Mn, Na, Zn, Al, B, Ca, Cd, Cr, Cu, Fe, K, Mg, Mn, Na, Ni, P, Pb, and Zn	Only manure resulted in highest contents of Total N, B, Cu, and Zn Manure + food waste resulted in highest contents of inorganic N, P, K, Ca, Fe, Mg, Mn, and Na Manure + organic waste resulted in lower fungal colonization and plant yield inhibition	[51]
Anaerobically digested sludge from water treatment	Cd, Cr, Cu, Ni, Pb, and Zn Total organic carbon N Available Ca, K, Mg, and P	Plant growth was increased Increased microbial biomass Cd and Zn immobilization in soil (low bioavailability) Reduced Cd uptake by plants AMF decreased Pb, Cd, Cu, and Zn bioavailability	[48]
Biomass resulted from phytoremediation with AMF	Cd and Zn	Cd and Zn reduced with AMF Methane produced 183 and 178 mL of CH_4_·g^−1^_VS_	[52]

**Table 10 biology-14-01579-t010:** List of articles with fungal strains and plants tested, as well as the reported effects on plant/fungi growth and pollutant/nutrient behaviors.

Fungi	Plant	Effects on Plant/Fungi Growth	Effects on Pollutant/Nutrient Behaviors	Reference
*Agaricus bisporus* and *Agaricus subrufescens*	N/A	*A. bisporus* biomass was 1.5 times higher than *A. subrufescens*	*A. subrufescens* decreased Ca, Na, Cr, Si, Mg, and Zn by 17–30%, and decreased Fe and Ni by 50–75% *A. bisporus* decreased Se, Si, and Co by 25–40% *A. bisporus* increased Mn and Se by 30% *A. subrufescens* increased B and Mo by more than 200%	[36]
*Agaricus bisporus* and *Agaricus subrufescens*	N/A	Not mentioned	Antibiotics were not accumulated. Removed 97% within the first days Anticonvulsant was removed faster by *Agaricus bisporus,* but with low uptake	[49]
*Agaricus bisporus* and *Agaricus subrufescens*	N/A	*A. subrufescens* grew 50% more than *A. bisporus* *Agaricus subrufescens* presented a 35% decrease in biomass yield between first and second harvests *Agaricus bisporus* presented a 50% decrease in biomass yield between first and second harvests	Low accumulation of PFAS in fungi *A. subrufescens* PFAS uptake: 14 ng/g (dry weight) *A. bisporus* PFAS uptake: 28 ng/g (dry weight)	[50]
*Pleurotus ostreatus* (non-mycorrhiza)	N/A	Manure alone produced 16 times more biomass than control and more than twice than manure + organic waste Lower fungi colonization and yield inhibition with manure + organic wastes	C/N changed from 32:1 to 29:1 after fruiting C/P changed from 189:1 to 186:1 after fruiting N/P remained the same after fruiting C/N ratios of 44:1 to 55:1 were most effective in terms of growth yield	[51]
AMF (*Claroideoglomus etunicatum*)	Hybrid *Pennisetum*	Plant growth achieved peak values @ highest digestate concentration (>13,216.67 kg/hm^2^)	Total nitrogen increase: 36% Nitrate nitrogen increase: 60% Ammonia nitrogen increase: 20% Available phosphorus increase: 50 times Soil organic matter decrease: 10%	[29]
AMF (*Rhizophagus irregularis*)	Maize (*Zea mays*)	Increase in plant yield: ca. 9%	Cd decrease: 0.77% Zn decrease: 0.13%	[52]
AMF (not strain specified)	Poplar	Plant growth was suggested, but not measured	Higher C, N, P contents when adding digestate No removal reported	[47]
AMF (*Funneliformis mosseae*)	Coriander (*Coriandrum sativum* L.)	Digestate without AMF increased plant growth by 100% With AMF, only 50% increase in plant growth	Pb decrease: 80% Cd decrease: 40% Zn decrease: 27% Cu remained the same Pb bioavailability increased in coriander with no digestate Pb bioavailability decreased with AMF and digestate Cu bioavailability increased by 50% without AMF	[48]
AMF (not strain specified)	Ryegrass (*Lolium perenne*)	Lower grass growth with digestate	Phosphate decrease: 30% Sulfate decrease: 75% Nitrate decrease: 100% Lower uptake of P, S, N, and K with digestate	[53]

**Table 11 biology-14-01579-t011:** Effect of digestate concentration on fungal diversity and colonization.

Digestate Concentration	AMF Diversity	Dominant Genera	Soil Fertility Effect	References
Low	Decreased	None	Limited	[47,48]
Moderate	Stable	*Glomus*	Improved	[29,47]
High	Increased	*Glomus*, *Paraglomus*	Strongly improved	[29,47,53]

**Table 12 biology-14-01579-t012:** Fungal strains and their tolerance to anaerobic digestate.

Fungal Strain	Growth Performance	Pollutant Tolerance	Key Accumulated Nutrients	References
*Pleurotus ostreatus*	High	Moderate	C, N, trace elements	[36]
*Lentinula edodes*	High	Moderate	Balanced C/N assimilation	[36]
*Pleurotus eryngii*	High	Moderate	C, N, Zn	[36]
*Agaricus bisporus*	Moderate	Higher PFAS uptake	K, Na	[36,49]
*Agaricus subrufescens*	High	Lower PFAS uptake	Cu, Zn	[36,49]

**Table 13 biology-14-01579-t013:** Contaminants and fungal mechanisms of removal/immobilization.

Contaminant	Fungal Strain(s)	Mechanism(s) Involved	Outcome	References
Cd, Zn	AMF, *Agaricus* spp.	Complexation, immobilization, bioaccumulation	Reduced plant uptake	[29,48]
Pb	AMF + digestate	Adsorption, glomalin, pH buffering	Reduced bioavailability	[48]
Cu	*A. subrufescens*	Bioaccumulation	Increased fungal content	[36]
Antibiotics (fluoroquinolones)	*A. subrufescens*	Photolysis, sorption, enzymatic degradation	High removal, low accumulation	[49]
Carbamazepine	*Agaricus* spp.	Sorption-dominated, limited enzymatic breakdown	Low removal	[49]

**Table 14 biology-14-01579-t014:** Soil and environmental factors affecting fungal efficiency.

Factor	Effect on Fungi/AMF	Consequences for Soil Remediation	References
C/N ratio	Low = improved colonization; High = N limitation	Higher/lower plant yield, fungal sporulation	[51,65]
Feedstock type	Manure = higher colonization; organic waste = inhibition	Plant yield and microbial stability	[51,60]
Conductivity/salinity	High = osmotic stress	Reduced fungal colonization	[51,60]
pH	Neutral favored fungal growth	TE immobilization, plant nutrition	[50]
Temperature	20 °C (*A. subrufescens*) vs. 18 °C (*A. bisporus*)	Contaminant removal efficiency	[50]

## Data Availability

The data presented in this study are available on request from the corresponding author due to privacy.

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
