# Peer review of "Use of Anaerobic Digestate Inoculated with Fungi as a Soil Amendment for Soil Remediation: A Systematic Review"

_biology, 2025, doi:10.3390/biology14111579_

Round 1

Reviewer 1 Report

Comments and Suggestions for Authors
  1. Lines 56–79 serve as the introduction section; it is unreasonable for such a large paragraph to lack literature citations.
  2. The formatting of the subscript at line 92 needs to be addressed.
  3. Check whether the beginning of line 93 has unintentionally omitted relevant sentences due to a clerical error and complete it accordingly.
  4. It is recommended to add literature citations for the four stages of anaerobic digestion mentioned at line 94 to confirm the accuracy of the description.
  5. Add a literature reference at line 97 to verify whether the statement "carbohydrates become sugars" is scientifically precise.
  6. At line 102, check whether the sentence is redundant with the previous one; it is suggested to revise the sentence to make it more concise.
  7. Evaluate whether the use of "acceptable" at line 110 is appropriate.
  8. Confirm the accuracy of the term used at line 122: are methanogens bacteria or archaea?
  9. The superscript position in the unit for CHâ‚„ productivity at line 130 needs adjustment. Additionally, the conclusion drawn in this passage is somewhat problematic, as the composition of biogas can indeed be influenced to increase the methane proportion through certain methods.
  10. The content in Table 2 should either be fully centered or fully left-aligned for consistency.
  11. Large sections of the text cite reference [3]; it is recommended to incorporate more recent high-quality literature to refine the article's content.
  12. Add literature citations in sequential order at line 170.
  13. Can [2] be used as the beginning of a sentence at line 176? The language should be revised.
  14. If the digestate no longer contains biodegradable substances, what is the reason for its role in promoting agricultural production? These issues require further detailed explanation (line 201).
  15. The writing at line 215 is typical of AI-generated content! If literature citations are also AI-generated, the credibility of the article will be significantly reduced.
  16. The full form of AMF (Arbuscular Mycorrhizal Fungi) is not provided upon its first occurrence.
  17. Table 6 summarizes the literature but lacks detailed data support.
  18. While temperature and pH are mentioned as key factors, the specific ranges or mechanisms of these parameters in the included studies are not systematically analyzed. It is recommended to supplement this in the article.
  19. The current conclusion does not clearly indicate how this review supplements existing knowledge or specific directions for future research. Strengthen the conclusion section to clarify the academic and practical value of the review.

Author Response

Answers to Reviewer 1:

Thanks a lot for the suggested corrections, they have given this paper a better scientific quality.

  1. Lines 56–79 serve as the introduction section; it is unreasonable for such a large paragraph to lack literature citations. Corrected.
  2. The formatting of the subscript at line 92 needs to be addressed. Corrected.
  3. Check whether the beginning of line 93 has unintentionally omitted relevant sentences due to a clerical error and complete it accordingly. Corrected.
  4. It is recommended to add literature citations for the four stages of anaerobic digestion mentioned at line 94 to confirm the accuracy of the description. Corrected..
  5. Add a literature reference at line 97 to verify whether the statement "carbohydrates become sugars" is scientifically precise. Corrected.
  6. At line 102, check whether the sentence is redundant with the previous one; it is suggested to revise the sentence to make it more concise. Corrected.
  7. Evaluate whether the use of "acceptable" at line 110 is appropriate. Corrected.
  8. Confirm the accuracy of the term used at line 122: are methanogens bacteria or archaea? Corrected.
  9. The superscript position in the unit for CHâ‚„ productivity at line 130 needs adjustment. Additionally, the conclusion drawn in this passage is somewhat problematic, as the composition of biogas can indeed be influenced to increase the methane proportion through certain methods. Corrected.
  10. The content in Table 2 should either be fully centered or fully left-aligned for consistency. Corrected.
  11. Large sections of the text cite reference [3]; it is recommended to incorporate more recent high-quality literature to refine the article's content. Corrected, updated with recent literature.
  12. Add literature citations in sequential order at line 170. References were mentioned above, references are listed in the order they are first cited in the text.
  13. Can [2] be used as the beginning of a sentence at line 176? The language should be revised. Corrected.
  14. If the digestate no longer contains biodegradable substances, what is the reason for its role in promoting agricultural production? These issues require further detailed explanation (line 201). Corrected.
  15. The writing at line 215 is typical of AI-generated content! If literature citations are also AI-generated, the credibility of the article will be significantly reduced. I did not use AI for my text, my word version is in German and the text you see appeared as “Error, reference source could not be found”.
  16. The full form of AMF (Arbuscular Mycorrhizal Fungi) is not provided upon its first occurrence. Corrected.
  17. Table 6 summarizes the literature but lacks detailed data support. I changed tables 7 – 10, to summarize better what I wanted to transmit, and added Tables 12 – 14.
  18. While temperature and pH are mentioned as key factors, the specific ranges or mechanisms of these parameters in the included studies are not systematically analyzed. It is recommended to supplement this in the article. Corrected, I as a result.
  19. The current conclusion does not clearly indicate how this review supplements existing knowledge or specific directions for future research. Strengthen the conclusion section to clarify the academic and practical value of the review. Corrected. 

Reviewer 2 Report

Comments and Suggestions for Authors

See attached files

Comments on the Quality of English Language

The manuscript requires revision to improve readability. Issues include long sentences, minor grammar errors (e.g., subject–verb agreement, missing articles), repetitive phrasing, and inconsistent use of terminology (e.g., “mycoremediation” vs. “fungal inoculation”). Some sections are wordy and could be streamlined for clarity. 

Author Response

Answers to Reviewer 2:

Thanks a lot for the suggested corrections, they have given this paper a better scientific quality.

The manuscript addresses the intersection of anaerobic digestion, digestate valorization, and mycoremediation. The integration of fungal inoculation with digestate application for soil remediation is scientifically compelling and relevant for circular bioeconomy, sustainable agriculture, and environmental biotechnology. The review highlights both the potential benefits (nutrient recycling, improved soil structure, pollutant attenuation) and challenges (nutrient overloading, heavy metal accumulation, ecological risks).  

However, several aspects could be strengthened to improve the paper’s scientific depth, readability, and impact. Specifically, the novelty of combining fungi–digestate interactions should be emphasized more strongly compared with existing literature on digestate-only or fungi-only remediation. Corrected, mentioned in chapter 1.4, 1.6, 2.3 and 4. Furthermore, some sections present descriptive content but lack critical synthesis of mechanisms, limitations, and research gaps. Corrected in the hole paper.

Specific Comments

  1. Abstract

The abstract is comprehensive, but the key quantitative findings (e.g., % reduction of specific pollutants, % improvements in soil nutrients) is limited. Corrected.

  1. Introduction

The novelty claim should be further strengthened: e.g., highlight why fungi–digestate interactions represent a distinct advancement beyond previous reviews of AD, digestate fertilization, or mycoremediation, etc. Corrected, mention in chapters 1 and 1.6.

  1. Results – Biomass Source of Digestate

Variability in C/N/P ratios of the digestate biomass source and their effect on fungi growth are limited, which could be further elaborated with mechanistic discussion (e.g., osmotic stress due to salts in food-waste digestate inhibiting mycelial colonization). Corrected.

Suggest to have a comparative table summarizing key digestate chemical parameters (pH, EC, nutrient content, heavy metals). Tables 7 – 10 were corrected and tables 12 – 14 added and discussed.

  1. Results – Fungal Strains
    • More synthesis is needed: Which strains show the highest tolerance to digestate-associated stressors (e.g., salinity, heavy metals)? Tables 7 – 10 were corrected and tables 12 – 14 added and discussed.
    • How does digestate concentration modulate fungal diversity and colonization efficiency? .
    •  

  1. Results – Pollutant/Nutrient Removal

The review covers metals (Cd, Zn, Pb, Cu), PFAS, and antibiotics. This diversity strengthens the paper. However, the discussion remains limited. A more indepeth discussion such as table or figure linking contaminants, fungal strains, mechanisms of removal/immobilization would be valuable. Tables 7 – 10 were corrected and tables 12 – 14 added and discussed.

  1. Plant Growth and Soil Health

The evidence for digestate improving soil structure and microbial biomass is well-summarized. However, conflicting results (e.g., digestate improving microbial diversity but sometimes reducing plant growth) need more critical discussion. Why do discrepancies occur (substrate type, digestate concentration, plant species, pH effects)? Discussion of organic matter removal could be linked more clearly to soil carbon cycling and long-term fertility. Corrected, mentioned in discussions.

  1. Others
  • The manuscript is generally well-written, but could benefit from improved conciseness (especially the Introduction and background on biogas). Corrected.
  • Ensure consistency in terminology: “mycoremediation,” “fungal inoculation,” and “mycorrhizal fungi” are sometimes used interchangeably but refer to distinct contexts.
  • Minor grammar and syntax issues should be polished (e.g., subject-verb agreement, overly long sentences). Corrected.

Round 2

Reviewer 2 Report

Comments and Suggestions for Authors

The revised manuscript has been well revised, and is recommended to be accepted for publication.